# FAD-TQ: Industrial Fine-grained Anomaly Detection with Thinking Quality

## Abstract

Recent research in industrial anomaly detection (IAD) has shifted beyond binary classification and segmentation, increasingly focusing on process-level, interpretable reasoning about the type and cause of anomalies. While multimodal large language models (MLLMs) have enabled this reformulation through visual question answering, current anomaly detection methods still suffer from two major limitations: the limited capacity of reward functions to capture intricate complexities and the reliance on generating supervised fine-tuning (SFT) data. Hence, we propose FAD-TQ, a lightweight reinforcement learning framework for fine-grained anomaly detection with thinking quality. Built upon the Group Policy Gradient paradigm, it eliminates the reference model and KL regularization to reduce rollout overhead and directly optimize the original reinforcement learning objective. To enable fine-grained guidance over the reasoning process, we design a thinking quality reward composed of two components: an efficiency reward that penalizes redundant reasoning, and a relevance reward that encourages task-aligned, coherent thought trajectories. Furthermore, we introduce MVTec-LOCO-AD-Pair3C, a principled evaluation protocol built on the existing dataset. By defining three decision types—normal, structural anomaly, and logical anomaly, rather than binary classification. Extensive experiments demonstrate that FAD-TQ improves interpretability, accuracy, streamlined reasoning and training efficiency with reduced computational costs. It demonstrates the potential of using small-scale benchmarks to evaluate MLLM capabilities in IAD. We hope this framework and evaluation protocol can serve as an example for future research on process-level reasoning in anomaly detection.

## 1 Introduction

Industrial anomaly detection (IAD) aims to identify rare defects and irregularities in manufacturing (Liu et al., 2024), playing a vital role in ensuring product quality and operational safety. Existing methods mainly fall into two paradigms: zero-shot and few-shot (Jeong et al., 2023). Zero-shot methods (Zhou et al., 2023; Cao et al., 2024) leverage anomaly data from seen categories and use prompt tuning to generalize to unseen object types and anomaly types, often without requiring fine-tuning on the test category. In contrast, few-shot methods (Chen et al., 2023; Li et al., 2024) build lightweight memory structures from a handful of normal samples within the same target domain, often combined with prompt adaptation. While effective, both zero-shot and few-shot approaches lack step-by-step, interpretable reasoning. To address this limitation, recent works reformulate anomaly detection as a visual question answering (VQA) problem using multimodal large language models (MLLMs) (Zhang et al., 2025; Li et al., 2023). These models produce natural language explanations alongside predictions, providing the first step toward interpretable anomaly detection. Building on this, reasoning-augmented MLLMs exemplified by OpenAI o1 (Jaech et al., 2024) and DeepSeek-R1 (Guo et al., 2025), and inspired by the Chain-of-Thought paradigm (Wei et al., 2022) explicitly generate intermediate reasoning steps before producing final answers, enabling step-by-step fine-grained reasoning. Despite the progress in reasoning-augmented IAD (Chao et al., 2025; Li et al., 2025a), existing methods remain constrained by three critical limitations. First, they suffer from reward insufficiency, where reinforcement learning only evaluates the final answer and format, offering no guidance on the intermediate reasoning quality. This weakens credit assignment and leads to inefficient training. Second, they rely on synthetically generated data for the initial Supervised

Fine-Tuning (SFT) phase (Li et al., 2025b; Zhao et al., 2025). Such data are expensive to construct and often domain-specific, limiting scalability. Third, their thinking processes are overlong which are not concise enough. These lead to a critical question: *Can we develop new approaches that harness the benefits of reasoning ability while addressing these limitations of data scale and computational efficiency?*

To address these limitations, we propose **FAD-TQ**, a lightweight reinforcement learning framework tailored for fine-grained anomaly detection with high-quality reasoning. First, we adopt the Group Policy Gradient (GPG) paradigm (Chu et al., 2025), which removes both the reference model and KL regularization. This simplification reduces GPU memory usage and rollout latency, while directly optimizing the original reinforcement learning objective without relying on auxiliary regularization terms. Second, to improve the quality of intermediate reasoning, we introduce a novel *thinking-quality reward* composed of two components: an *efficiency reward* that discourages redundant or verbose reasoning, and a *relevance reward* that promotes task-relevant thinking behaviors aligned with the ground truth. By jointly constraining conciseness and logical relevance, this reward encourages the model to generate reasoning processes that are not only faithful to the task but also streamlined and efficient. Third, we introduce **MVTec-LOCO-AD-Pair3C**, a structured evaluation protocol built on top of the existing MVTec LOCO dataset (Bergmann et al., 2022). By defining three decision types—normal, structural anomaly, and logical anomaly— this protocol supports efficient training and evaluation under limited data. Although MVTec-LOCO-AD-Pair3C is relatively small in scale compared to MMAD (Jiang et al., 2025), it remains a principled and representative benchmark for IAD tasks. Its lightweight design makes it a practical alternative for scenarios where large-scale benchmarks like MMAD are prohibitively time-consuming to use, while still offering meaningful evaluation for MLLMs.

In summary, our contributions are three-fold:

- We develop **FAD-TQ**, a lightweight and scalable reinforcement learning framework based on the Group Policy Gradient (GPG) paradigm, which eliminates the reference model and KL regularization, thereby directly optimizing the original RL objective while substantially reducing memory footprint and rollout latency.
- We design a novel *thinking-quality reward* that jointly evaluates the conciseness and task relevance of reasoning trajectories, encouraging the model to generate shorter yet more effective reasoning chains. This provides fine-grained credit assignment and enhances both interpretability and fidelity of model decisions.
- We propose **MVTec-LOCO-AD-Pair3C**, a principled and low-cost evaluation protocol that defines three decision types and serves as a representative lightweight alternative to MMAD. This protocol lowers the barrier to future research by enabling fast experimentation and fair comparison without relying on large-scale synthetic datasets.

Our code and implementation details are partly open-sourced here.

## 2 APPROACH

### 2.1 TASK DEFINITION

We recast industrial anomaly detection as a visual–language reasoning problem, a MLLM is expected to produce a natural-language verdict instead of conventional anomaly logits. Each instance consists of a reference image $I_r \in \mathbb{R}^{H \times W \times 3}$ representing a normal sample from a specific object category, and a test image $I_t \in \mathbb{R}^{H \times W \times 3}$ whose anomaly status is to be determined. The model is tasked with determining whether the test image $I_t$ is: (1) Normal (no anomaly), (2) Structurally anomalous (e.g., missing parts, deformation, or defects), (3) Logically anomalous (e.g., count mismatch or spatial inconsistency). Formally, we denote the multimodal model as $\pi_\theta$, parameterized by $\theta$. Given the input image pair $(I_r, I_t)$ and the task prompt $x$, the model outputs a class label:

$$\pi_\theta(I_r, I_t, x) \to y \in \{y_{\text{normal}}, y_{\text{struct}}, y_{\text{logical}}\}.$$

Only $y$ is used for metrics computation. The reference $I_r$ is guaranteed normal and sampled from the same category as $I_t$.

Table 1: Dataset statistics of MVTec-LOCO-AD-Pair3C showing counts for Good samples, Structural Anomalies (SA), and Logical Anomalies (LA), partitioned by training and test splits.

| Category | Total | | Good | | SA | | LA | |
|---|---|---|---|---|---|---|---|---|
| | Train | Test | Train | Test | Train | Test | Train | Test |
| Breakfast Box | 120 | 155 | 40 | 62 | 40 | 50 | 40 | 43 |
| Juice Bottle | 120 | 210 | 40 | 54 | 40 | 54 | 40 | 102 |
| Pushpins | 120 | 190 | 40 | 98 | 40 | 41 | 40 | 51 |
| Screw Bag | 120 | 221 | 40 | 82 | 40 | 42 | 40 | 97 |
| Splicing Connectors | 120 | 192 | 40 | 79 | 40 | 45 | 40 | 68 |
| **Total** | **600** | **968** | **200** | **375** | **200** | **232** | **200** | **361** |

## 2.2 BENCHMARK: MVTEC-LOCO-AD-PAIR3C

We introduce **MVTec-LOCO-AD-Pair3C**, a lightweight yet fine-grained benchmark designed to evaluate multimodal reasoning capabilities for industrial anomaly detection.

**Source Dataset.** Our benchmark is built upon the MVTec LOCO AD dataset, which includes five industrial object categories: *breakfast box*, *juice bottle*, *pushpins*, *screw bag*, and *splicing connectors*. Each category provides training set, a validation set containing only normal samples, and test setcontaining three fine-grained sub-classes: *normal*, *structural anomaly* (e.g., missing parts, surface defects), and *logical anomaly* (e.g., incorrect object counts or spatial misconfigurations).

**Balanced Training–Test Split.** To construct a small yet representative training set, we perform stratified sampling on the official test split. Specifically, for each object category and each sub-class (i.e., *Good*, *Structural anomalies*, *Logical anomalies*), we randomly sample 40 images as the training subset. The remaining images in each sub-class are reserved for evaluation. To adapt this data to our pairwise setting, we generate ref–test pairs as follows: In the training set, each test image $I_t$ is paired with a reference image $I_r$ sampled from the official training split (normal only). In the test set, each test image $I_t$ is paired with a reference image $I_r$ sampled from the official validation split (normal only). This protocol guarantees: (i) balanced coverage of all anomaly types in training; (ii) strict disjointness between training and test images; (iii) no additional annotation cost, as all labels are inherited from the original dataset. More details are shown in 1.

**Pairwise Classification Protocol.** We formulate the task as a 3-way classification over image pairs. Each instance is represented as a pair $(I_r, I_t)$: For training, $I_r$ is randomly sampled from the original training set (normal only). For testing, $I_r$ is randomly sampled from the original validation set (normal only). Given a pair $(I_r, I_t)$, the model predicts a class label $y \in \{good, structural, logical\}$. This extends the original binary anomaly detection setup into a more challenging, interpretable, and fine-grained 3-class reasoning task, as shown in 1.

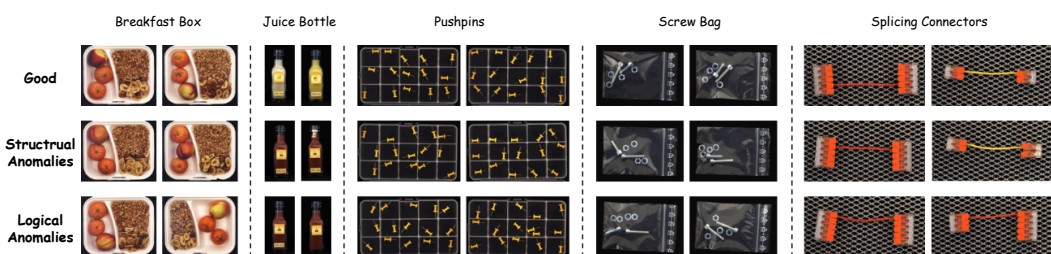

Figure 1: Example samples from the MVTec-LOCO-AD-Pair3C benchmark. Each column corresponds to a distinct object category, while each row represents a different anomaly type: good (top), structural anomaly (middle), and logical anomaly (bottom). This visualization highlights the fine-grained anomaly detection rather than binary classification.

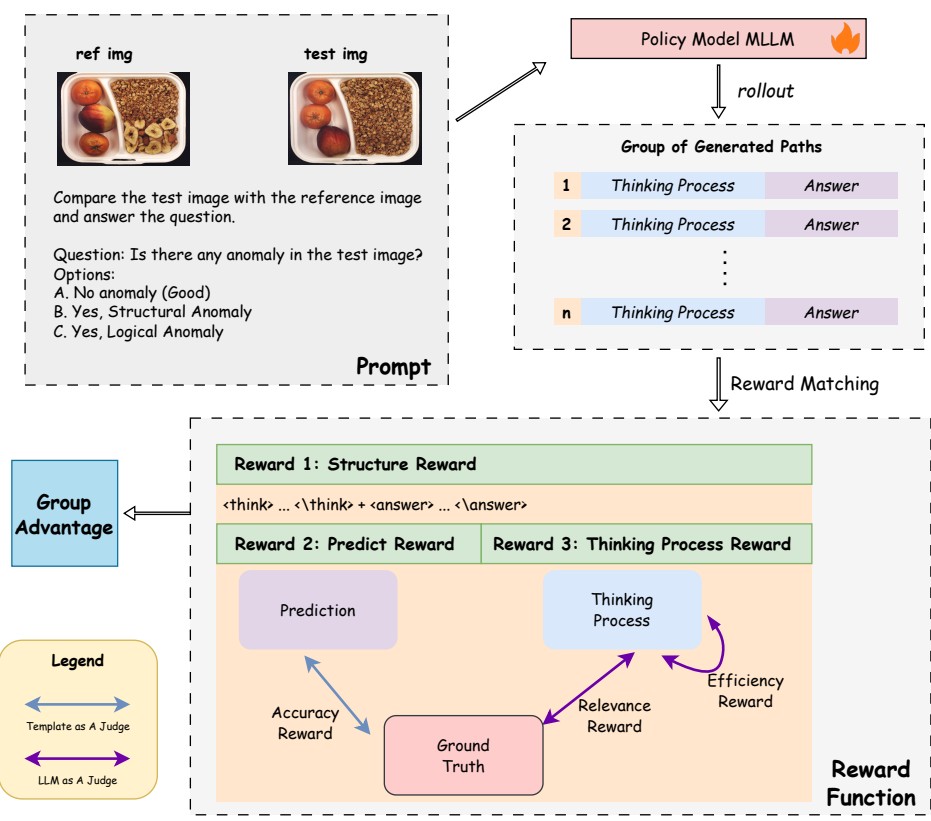

Figure 2: Overview of our proposed pipeline FAD-TQ. Given a reference–test image pair and a multiple-choice question, the policy model samples a group of reasoning trajectories. Each trajectory is assigned a composite reward based on structural coherence, answer correctness, and thinking quality. This thinking qulity reward is derived from a dual-judge mechanism, as shown in the legend: the final prediction's accuracy is verified programmatically via template matching against the ground truth (blue arrow), while the thinking process is evaluated semantically by an LLM-as-a-Judge on its quality and efficiency (purple arrows). Advantages are then computed by normalizing these rewards within the group.

## 2.3 METHOD

We introduce Fine-grained Anomaly Detection with Thinking Quality (**FAD-TQ**), a lightweight group-based RL framework. Given $(I_r, I_t, q)$, the policy samples a group of reasoning trajectories, each containing a thinking process and a final answer. A process-aware reward evaluates format compliance, prediction accuracy, and thinking quality (efficiency and task relevance), producing a scalar return per trajectory. Returns are normalized within the group to form advantages, and the policy is updated by a clipped objective without a reference model or KL regularization. Figure 2 provides an overview of the framework.

**Group Policy Gradient Paradigm.** For policy optimization, we adopt the Group Policy Gradient (GPG) algorithm, a critic-free method that stabilizes learning by leveraging batch-level statistics instead of a learned value function. Specifically, for a group of $N$ trajectories sampled from the current policy, the advantage $A(\tau_i)$ of a trajectory is computed by standardizing its return $R(\tau_i)$ against the group's empirical mean ($\mu$) and standard deviation ($\sigma$):

$$A(\tau_i) = \frac{R(\tau_i) - \mu}{\sigma + \varepsilon}$$

where $\varepsilon$ is a small constant for numerical stability.

The policy parameters $\theta$ are then updated by ascending the gradient of the standard policy gradient objective, weighted by this advantage:

$$\mathcal{L}_{GPG}(\theta) = \mathbb{E}_{\tau \sim \pi_\theta}[\log \pi_\theta(\tau) A(\tau)]$$

Crucially, GPG's formulation is deliberately simpler than that of proximal methods like GRPO. By forgoing a reference model and a surrogate objective, it avoids the computational overhead associated with importance sampling and directly optimizes the fundamental policy gradient objective.

**Reward Function Definition.**    To enable fine-grained reasoning in anomaly detection, we define three types of scalar rewards: (1) *format reward* that enforces response structure, (2) *accuracy reward* that evaluates the final prediction correctness, and (3) *thinking quality reward* that supervises the reasoning process. Each trajectory is scored by a weighted combination of these rewards, enabling fine-grained credit assignment across the reasoning trajectory. We describe each reward component in detail below.

(1) **Format Reward and Accuracy Reward.** To ensure structural consistency in model responses, we define a *format reward* and an *accuracy reward*. The *format reward* $r_{\text{format}}(\tau)$ checks whether the model output strictly matches the required response format: a reasoning segment enclosed in *think* tags followed by a final answer enclosed in *answer* tags. If both components are present and properly ordered, the reward is 1; otherwise, it is 0:

$$r_{\text{format}}(\tau) = \begin{cases} 1, & \text{if output matches } \textit{<think>...</think><answer>...</answer>} \\ 0, & \text{otherwise} \end{cases}$$

The *accuracy reward* $r_{\text{accuracy}}(\tau)$ compares the extracted answer span with the ground-truth class label. A reward of 1 is assigned if the predicted choice matches the ground truth, and 0 otherwise:

$$r_{\text{accuracy}}(\tau) = \begin{cases} 1, & \text{if predicted label matches ground truth} \\ 0, & \text{otherwise} \end{cases}$$

(2) **Thinking-Quality Reward.** To provide fine-grained supervision over the intermediate reasoning process, we design a *thinking-quality reward* $r_{\text{tq}}(\tau)$ based on LLM evaluation. Specifically, we extract the reasoning text enclosed by *think* tags from each trajectory $\tau$, together with the predicted answer from the *answer* segment. A frozen LLM is then used to assess reasoning quality along two complementary dimensions:

- **Conciseness:** whether the reasoning avoids redundancy and unnecessary elaboration. A frozen LLM judge scores this dimension on a 0–3 scale, with higher scores indicating shorter, clearer, and more efficient reasoning.
- **Relevance (Logical Relevance):** whether the reasoning logically supports the final answer and explicitly compares the reference and test images. A frozen LLM judge also scores this on a 0–3 scale, reflecting coherence, structure, and visual grounding of the reasoning.

The scores are extracted and normalized to the range $[0, 1]$:

$$r_{\text{conc}}(\tau), \ r_{\text{rel}}(\tau) \in [0, 1].$$

The final reward is a weighted combination of the two components:

$$r_{\text{tq}}(\tau) = \eta \cdot r_{\text{conc}}(\tau) + (1 - \eta) \cdot r_{\text{rel}}(\tau), \quad \eta \in [0, 1],$$

where $\eta$ balances the importance of conciseness and relevance. This design captures both structural efficiency and semantic faithfulness of the reasoning trace, providing informative training signals even when the final prediction is correct but the reasoning is suboptimal.

In summary, the final reward for each trajectory $\tau$ is computed as:

$$R(\tau) = r_{\text{format}}(\tau) + r_{\text{accuracy}}(\tau) + \min\big(1, \ \max(0, \ r_{\text{accuracy}}(\tau))\big) \cdot r_{\text{tq}}(\tau)$$

where each component returns a scalar in $[0, 1]$, and the total reward satisfies $R(\tau) \in [0, 3]$. The $r_{\text{tq}}$ component is only activated when the predicted answer is correct (i.e., accuracy reward equals 1).

## 3 EXPERIMENTS

### 3.1 SETUP

**Datasets and Metrics.** We conduct experiments on the MVTec-LOCO-AD-Pair3C benchmark described in Section 2.2. The benchmark transforms industrial anomaly detection into a challenging multimodal reasoning task with 3-way classification (*good*, *structural*, *logical*) over reference-test image pairs. Our training set contains 600 sampled pairs (40 per class across 5 categories), while the test set comprises the remaining images from each category, totaling approximately 2,400 evaluation pairs. This balanced yet compact design enables efficient experimentation while maintaining comprehensive coverage of industrial anomaly types.

We adopt **Accuracy** as the primary evaluation metric. For a three-class classification task with test pairs $\{(I_ref^{(i)}, I_test^{(i)}, y^{(i)})\}_{i=1}^N$ and predictions $\hat{y}^{(i)}$, accuracy is defined as:

$$\text{Accuracy} = \frac{1}{N} \sum_{i=1}^{N} \mathbf{1}\big[\hat{y}^{(i)} = y^{(i)}\big].$$

This metric is chosen for its straightforward interpretability and its appropriateness for the balanced nature of our test set, providing a clear measure of the model's overall performance in distinguishing between normal, structural, and logical conditions. However, accuracy alone does not characterize the distribution of the model's outputs. For example, a model could achieve a non-trivial accuracy score by consistently selecting a single, high-frequency option.

To provide a more complete picture of model behavior at the option level, we introduce a complementary diagnostic metric named **Balance Score**. The premise is straightforward: given that our benchmark is designed with a balanced distribution of correct answers across the three classes, a well-behaved model should not produce an extremely skewed distribution of predictions We calculate this by first computing the probability distribution of the model's chosen options, $p = (p_{\text{good}}, p_{\text{structural}}, p_{\text{logical}})$. The score is the Shannon entropy of this distribution, normalized by the maximum possible entropy, $\log(3)$:

$$\text{Balance Score} = \frac{H(p)}{\log(3)}, \quad \text{where} \quad H(p) = -\sum_c p_c \cdot \log(p_c + \varepsilon).$$

This score lies in the range $[0, 1]$. A score approaching 0 indicates a degenerate output pattern where the model almost exclusively selects a single option. This metric is not intended as an optimization target. Instead, it serves a diagnostic purpose: An extremely low score flags that the model's response behavior is pathologically low variance. It helps us verify that a high accuracy score is not merely an artifact of a simplistic response bias. When reported alongside accuracy, the Balance Score offers crucial insight into the diversity of a model's outputs, ensuring a more rigorous and transparent evaluation.

**Implementation Details** We adopt Qwen2.5-VL-3B as the base model and train it using reinforcement learning algorithms provided by the verl (Sheng et al., 2025) framework [1], which provides support for reinforcement learning algorithms. We train the model for 10 epochs on the training set. Each training batch samples 32 prompts, with 8 rollouts generated per prompt. All experiments are run on 5 compute units, each equipped with 48GB of memory. We fix the random seed across all experiments and keep hyperparameters consistent across folds and comparison settings. For LLM as a judge, we take Qwen3-8B by vllm (Kwon et al., 2023) project[2] to check models' responses. $\eta$ is set to 0.5, and $\varepsilon$ is set to $10^{-12}$.

### 3.2 MAIN RESULTS

To provide a comprehensive evaluation, we compare our method against two primary architectural paradigms: dense models and Mixture-of-Experts (MoE) models. For dense models, we evaluate the Qwen2.5-VL series (3B to 32B) and MiniCPM-V-4 (Yao et al., 2024). As shown in Table 2, performance generally scales with model size, with Qwen2.5-VL-32B achieving the strongest overall

---

[1] https://github.com/volcengine/verl
[2] https://github.com/vllm-project/vllm

Table 2: Comparison of different models on various tasks. We report two metrics: accuracy (Acc.) and Balance Score (BS), both in percentage terms. For the accuracy metric, the best result is shown in **bold**, and the second-best is underlined. Balance scores are shown in gray to indicate that they serve as auxiliary indicators. Among them, scores below 15% are displayed in lighter gray to highlight the presence of prediction bias or degenerate option selection.

| Model | Type | Breakfast Box | | Splicing | | Juice Bottle | | Screw Bag | | Pushpins | | Overall |
| --- | --- | --- | --- | --- | --- | --- | --- | --- | --- | --- | --- | --- |
| | | Acc. | BS | Acc. | BS | Acc. | BS | Acc. | BS | Acc. | BS | |
| Kimi-VL-A3B-Instruct | MoE | 41.94 | 29.68 | 45.83 | 27.88 | 32.38 | 65.42 | 40.72 | 16.79 | 51.58 | 17.36 | 42.25 |
| Kimi-VL-A3B-Thinking | | 39.35 | 6.28 | 40.10 | 9.22 | 41.43 | 60.03 | 35.75 | 65.62 | 38.95 | 36.34 | 39.05 |
| Gemma-3n-E4B-it | | 41.29 | 70.50 | 35.42 | 54.08 | 47.62 | 60.83 | 40.72 | 50.34 | 39.47 | 63.09 | 41.01 |
| MiniCPM-V-4 | Dense | 40.65 | 14.44 | 41.15 | 0.00 | 25.71 | 0.00 | 37.10 | 0.00 | 51.58 | 11.08 | 38.84 |
| Qwen2.5-VL-3B | | 42.58 | 10.91 | 41.67 | 5.27 | 25.71 | 8.58 | 37.56 | 2.63 | 53.68 | 12.28 | 39.77 |
| Qwen2.5-VL-7B | | 46.45 | 38.94 | 46.35 | 44.74 | 27.62 | 9.79 | 42.08 | 49.98 | 55.26 | 14.36 | 43.08 |
| Qwen2.5-VL-32B | | 60.00 | 99.16 | 44.27 | 74.58 | 47.62 | 87.60 | 42.08 | 87.02 | 48.95 | 81.82 | 47.93 |
| FAD-TQ (Ours) | | **72.26** | 78.13 | **54.17** | 99.73 | **54.29** | 90.69 | **44.80** | 86.85 | 58.95 | 29.59 | **55.89** |

baseline score of 47.93. However, smaller dense models such as Qwen2.5-VL-3B, despite achieving reasonable accuracy, show much weaker balance performance. For Mixture-of-Experts (MoE) models, we evaluate Kimi-VL-A3B (Team et al., 2025b) and Gemma-3n-E4B (Team et al., 2025a), which demonstrate strong parameter efficiency. These models achieve performance competitive with dense models of a comparable activated parameter scale, and in general maintain higher balance levels than the smaller dense models. This suggests that sparse activation mitigates choices bias and encourages more robust decision-making even at smaller scales. Beyond raw accuracy, the Balance Score provides an important diagnostic perspective on model behavior. As highlighted by the gray-shaded entries in Table 2, some models achieve superficially reasonable accuracy while producing extremely low entropy across their predictions. In such cases, the model overwhelmingly favors a single option, revealing a degenerate output bias that undermines genuine reasoning ability. This underlines the necessity of reporting accuracy together with Balance Score, ensuring that improvements reflect balanced and robust reasoning rather than statistical shortcuts. Importantly, our **FAD-TQ** achieves the highest average accuracy of 55.89, representing improvements of approximately +13 points over Qwen2.5-VL-3B and +8 points over Qwen2.5-VL-32B. Performance gains are consistent across categories, such as breakfast box and splicing connectors. Moreover, the balance scores remain high, implying that the improvements do not come from biased or degenerate predictions. This combination of stronger accuracy and balanced reasoning behavior underscores the efficacy of our approach.

## 3.3 ABLATION STUDY

**Ablation of Effect of Training Strategies** We begin by assessing the effect of training strategies on the MVTec-LOCO-AD-Pair3C benchmark via a controlled ablation across four configurations: (1) the base model, Qwen2.5-VL-3B, without reinforcement learning, (2) the base model fine-tuned with GRPO, (3) the base model trained using GPG, and (4) our full method FAD-TQ. We evaluate both overall accuracy and the average reasoning length.

Table 3: Ablation on training strategies. We evaluate the effect of reinforcement learning methods (GRPO, GPG) and our full pipeline (FAD-TQ). Metrics include accuracy and average reasoning length. Ref is used here as an abbreviation for 'reference model'.

| Method | +Ref | Accuracy Gain | Avg. Length |
| --- | --- | --- | --- |
| Qwen2.5-VL 3B | | 39.77 | N/A |
| + GRPO | ✓ | +8.99 | 52.02 |
| + GPG | ✗ | +13.64 | 44.04 |
| **FAD-TQ (Ours)** | ✗ | **55.89** | **36.76** |

length. As shown in Table 3, FAD-TQ achieves the best accuracy while maintaining the shortest reasoning length, indicating that our method effectively balances correctness and efficiency. We observe that compared to GRPO and GPG, our method avoids unnecessarily verbose reasoning steps, indicating improved reasoning efficiency. This suggests that our reward formulation not only enhances output correctness, but also encourages the model to generate more concise and purposeful reasoning trajectories. Moreover, prior works on IAD reasoning MLLMs (Chao et al., 2025; Li et al., 2025b) typically exhibit much longer reasoning traces in their reported examples. In contrast,

Figure 3: Visualization of the results produced by FAD-TQ. For each class in the MVTec-LOCO-AD-Pair3C, a representative test case is presented, including a reference image, a query image, and a prompt inquiring whether an anomaly is present. The model's output is composed of a thinking process and the final answer.

our method encourages more concise and purposeful trajectories, avoiding unnecessarily lengthy deliberations while maintaining effectiveness.

**Ablation of Thinking Components Comparison.** Table 4 reports the contribution of each sub-reward in the thinking-quality design. Removing either efficiency or relevance reward consistently decreases accuracy, confirming that both conciseness and task-relevance are necessary for effective reasoning supervision. Fur-

Table 4: Impact of thinking quality reward components in FAD-TQ.

| Model | Accuracy |
|---|---|
| FAD-TQ (Full Model) | 55.89 |
| – Efficiency Reward | -4.10 |
| – Relevance Reward | -2.27 |

thermore, the efficiency reward is explicitly designed to penalize redundant or verbose reasoning, which regularizes the reasoning trajectory length and is expected to reduce unnecessary tokens during inference. This design encourages concise yet effective reasoning, contributing to better resource efficiency without sacrificing output quality. From the optimization perspective, the stability of policy updates is closely related to the reward distribution. When only a single sub-reward is applied, the reward variance is larger, which weakens the effectiveness of advantage estimation. Among the two, the relevance reward exhibits a relatively smoother distribution, which explains why it performs better than the efficiency reward when used alone. When both rewards are combined, normalization produces a more balanced and stable reward distribution, leading to smoother training dynamics and more reliable gradient signals. This highlights the importance of designing complementary sub-rewards rather than relying on a single dimension of reasoning quality.

## 4 RELATED WORK

**Reinforcement Learning for Reasoning in Large Models.** Reinforcement learning (RL) (Shao et al., 2024) has become a key approach for enhancing reasoning in large language models (LLMs) (Grattafiori et al., 2024) and multimodal LLMs (MLLMs) (Bai et al., 2025), moving beyond prompt engineering such as Chain-of-Thought (CoT) (Wei et al., 2022) to reward-driven fine-tuning. Group Relative Policy Optimization (GRPO) (Guo et al., 2025) replaces the value-based critic in PPO (Schulman et al., 2017) with group-based reward normalization, enabling stable RL without heavy overhead. It serves as a strong baseline for CoT-style reasoning tasks. However, GRPO suffers from instability in long sequences and reliance on reference models. Several methods address these issues with progressively improved designs. DAPO (Yu et al., 2025) introduces

decoupled clipping, dynamic sampling, and overlength reward shaping to enhance stability and convergence in long-form reasoning. GPG (Chu et al., 2025) eliminates both the critic and reference model, directly optimizing the policy gradient with group-based, bias-corrected advantage estimation, enabling fully reference-free training. GSPO (Zheng et al., 2025) redefines the objective at the sequence level using likelihood-based importance weights and clipping. This significantly improves robustness, especially for large-scale or Mixture-of-Experts models, where token-level gradients can be noisy. Building on this line, our method introduces a reference-free RL framework that leverages a reward model to guide multi-step reasoning, without relying on any critic or reference policy. This design enables scalable, low-cost fine-tuning that generalizes well to new models and tasks, especially in low-resource settings.

**Anomaly Detection with Reasoning MLLM.** A growing line of research investigates how to equip vision-language models with reasoning capabilities for industrial anomaly detection (Bergmann et al., 2019), particularly in scenarios with limited labeled data and complex, logical anomaly patterns. AnomalyR1 (Chao et al., 2025) is an early attempt to integrate GRPO-based reinforcement learning with a MLLM, using the Reasoned Outcome Alignment Metric (ROAM) to guide precise anomaly localization and segmentation in data-scarce settings. LAD-Reasoner (Li et al., 2025a) leverages a small-scale MLLM (Qwen2.5-VL 3B) and introduces a two-stage pipeline: supervised fine-tuning (SFT) for basic visual understanding, followed by GRPO-driven reinforcement learning for logical anomaly detection with interpretable reasoning steps. OmniAD (Zhao et al., 2025) formulates anomaly detection as a unified multimodal reasoning problem. It combines text-as-mask encoding and visual-guided textual reasoning, trained via a hybrid SFT and GRPO procedure, achieving strong few-shot performance through sophisticated reward design. IAD-R1 (Li et al., 2025b) proposes a universal post-training framework combining perception-activated SFT and structured-control GRPO (SC-GRPO), optimized with multi-dimensional rewards to enhance anomaly interpretation and reasoning consistency. Together, these methods highlight the potential of reinforcement learning for reasoning in industrial anomaly detection. However, these methods often depend on expensive data generation. Our method bypasses this cost by training directly on existing tasks without additional data construction, achieving comparative performance with minimal supervision.

## 5 CONCLUSION

We revisit industrial anomaly detection from a reasoning-centric perspective, and propose a fine-grained formulation named MVTec-LOCO-AD-Pair3C, which moves beyond binary anomaly detection between normal and abnormal samples to additionally identify anomaly types such as structural or logical defects. To support efficient evaluation, we construct a benchmark derived from MVTec-LOCO, serving as an efficient and practical choice for evaluation while requiring significantly fewer computational resources. To tackle this task, we introduce **FAD-TQ**, a reinforcement learning framework built upon the GPG paradigm. Our method removes the reference model and KL regularization, yielding a principled simplification that reduces memory and training overhead. Moreover, we design a novel reward that supervises the *thinking process*, encouraging conciseness and logical relevance in reasoning by penalizing redundancy and rewarding task-grounded outputs. Extensive experiments demonstrate that FAD-TQ achieves competitive performance under designed settings. Importantly, the approach can consistently yield performance gains and demonstrates broad compatibility with diverse and evolving MLLMs, underscoring its potential as a general paradigm for reasoning-enhanced anomaly detection.

## 6 STATEMENT OF LLM USAGE

During the preparation of this manuscript, we utilized LLMs to assist with grammar correction, phrasing refinement, and overall readability. All suggestions provided by the LLM were carefully reviewed, edited, and revised by the authors, and the authors retain full responsibility for the final version of this paper.

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
