# OpenReview forum: "FAD-TQ: Industrial Fine-grained Anomaly Detection with Thinking Quality"
_ICLR.cc/2026/Conference — Submitted to ICLR 2026_

### Official Review · Reviewer_41iH · 2025-10-15

**Soundness:** 2
**Presentation:** 3
**Contribution:** 2
**Rating:** 4
**Confidence:** 4

**Summary:**

This paper aims at achieving better reasoining capability of image anomaly detection. To achieve this, it split the MVTec-LOCO dataset with logical and structure anomalies to set up a anomaly-type-classified testbed, and adopt Group Policy Gradient paradigm with two reward setup to penalizes redundant reasoning and coherence. The proposed methods achieves better performance on the proposed benchmark.

**Strengths:**

1) This paper achieves better reasoning capability for image anomalies by adopting the Group Policy Gradient paradigm for MLLM post-training.

2) The decision to split anomaly categories is a sound approach for rigorously evaluating the model's fine-grained classification capabilities.

3) The proposed empirical reward designs are effective, according to the qualitative results presented.

**Weaknesses:**

My primary concern with this paper is the insufficient comparison to relevant baselines and the absence of evaluation on a much larger, established benchmark for anomaly reasoning, namely MMAD.

1) The paper's comparison is limited to several zero-shot LLMs. It lacks a comprehensive evaluation against a wider scope of multi-modal LLMs that have been specifically designed or fine-tuned for anomaly detection and reasoning tasks.

2) The core of the proposed method involves leveraging the Group Policy Gradient paradigm, a technique extensively studied in reinforcement learning and widely applied to LLM post-training. Applying this existing technique to the narrow domain of anomaly detection, without significant novel adaptation, makes the paper's contribution appear marginal.

3)  A comprehensive benchmark for multi-modal anomaly detection and reasoning, MMAD, already exists. It is strongly recommended that the authors evaluate their proposed method on this benchmark. A strong performance on a large-scale, recognized dataset like MMAD would provide a much more robust validation of the method's capabilities than the results on the small-scale, curated dataset currently used.

**Questions:**

Could you please clarify the classification protocol for samples in the MVTec-LOCO dataset that exhibit both structural and logical anomalies? Table 1 suggests that these two categories are mutually exclusive. It is unclear how cases with multiple anomaly types are handled—are they assigned to a single, dominant type, or categorized in another way?

---

> ### Author Response · Authors · 2025-11-24
>
> We thank the reviewer for their constructive critique and the strong recommendation to include the MMAD benchmark. We have addressed the concerns as follows:
>
> **Q1:** Evaluation on the MMAD Benchmark.
>
> **A1:** We appreciate the reviewer pointing out the necessity of evaluating on a larger, established benchmark. We strictly followed this advice and extended our evaluation to the **MMAD benchmark**, a large-scale, multi-domain dataset, to validate the robustness of our method.
>
> We compared our method against the Base model and a standard SFT counterpart (representing the standard fine-tuning paradigm) under the same setting. The results are presented in **Table 1**.
>
> Table 1. Evaluation on OOD Benchmark (MMAD). MMAD is evaluated under a one-shot setting to test generalization capabilities.
>
> | Model | FAD Acc (Target) | MMAD Avg | MMAD Prec | MMAD Recall | MMAD F1 |
> | :--- | :---: | :---: | :---: | :---: | :---: |
> | Baseline (Qwen2.5-VL-3B) | 39.77 | 63.3 | **72.3** | 34.5 | 40.8 |
> | SFT | 48.97 | 36.4 | 11.5 | 7.3 | 8.8 |
> | **Ours** | **55.89** | **63.9** | 71.9 | **40.3** | **44.7** |
>
> **Analysis:**
> The results on MMAD provide robust validation of our method and highlight a critical insight regarding baselines:
> 1.  **Superior Generalization:** Our method achieves an F1 score of **44.7%** and Recall of **40.3%**, significantly outperforming standard SFT (F1 8.8%, Recall 7.3%).
> 2.  **Addressing SFT Limitations:** The comparison highlights that standard supervised fine-tuning—a common approach for adapting MLLMs—suffers from severe **catastrophic forgetting** on large-scale benchmarks. Our GPG-based approach effectively aligns the model without sacrificing its inherent generalization capabilities.
> 3.  **Scale and Robustness:** By validating on MMAD, we confirm that our method is not limited to small-scale curated datasets but scales effectively to diverse industrial scenarios.
>
> **Q2:** Novelty and Contribution.
>
> **A2:** We respectfully clarify that our contribution is not merely applying the existing GPG technique, but rather **identifying and solving the fundamental failure modes of standard SFT in the IAD domain** through a tailored framework.
>
> 1.  **Solving "Catastrophic Forgetting" in IAD:** As evidenced by our new MMAD results (Table 1), directly applying standard fine-tuning (SFT) to anomaly detection leads to a collapse in generalization (F1 dropping to 8.8%). Our novelty lies in demonstrating that the GPG paradigm is the **necessary solution** to preserve MLLM's reasoning capabilities while adapting to industrial needs, a finding that challenges the prevalence of SFT in this domain.
> 2.  **Task-Specific Reward Engineering:** We introduced the **Thinking Quality Reward** specifically to tackle the "verbosity" and "hallucination" issues common in MLLMs when describing anomalies. This is not a generic RL application but a domain-specific adaptation that forces the model to generate concise, evidence-based diagnoses, which standard generic RLHF does not address.
>
> **Q3:** Clarification on the classification protocol.
>
> **A3:** We confirm that the classifications presented in Table 1 for our evaluation are strictly mutually exclusive.
>
> 1.  **Dataset Definition:** In the official MVTec-LOCO benchmark, each test sample is assigned a **single, unique ground-truth label**. The samples are organized into distinct subdirectories (e.g., `logical_anomalies` vs. `structural_anomalies`), and there are **no multi-label cases** in the standard test set where a single image is annotated as containing both types simultaneously.
> 2.  **No Ambiguity:** Consequently, there is no fuzzy classification or overlap in the ground truth. Our experimental setup strictly adheres to this official protocol. Each sample corresponds to exactly one category, and we evaluate our model's accuracy based on this determinant mapping.

---

> > ### Comment · Reviewer_41iH · 2025-11-27
> >
> > Thank you for the response. However, I still find the novelty and contribution of this work insufficient. After checking the reviews and author feedbacks of our reviewers, I will maintain my score.

---

### Official Review · Reviewer_T6EL · 2025-10-31

**Soundness:** 3
**Presentation:** 4
**Contribution:** 2
**Rating:** 4
**Confidence:** 5

**Summary:**

This paper addresses efficiency issues in multimodal large language model-based Anomaly Detection (MLLM-AD) by proposing FAD-TQ. It removes the reference model and KL regularization to reduce computational overhead and introduces a thinking-quality reward combining efficiency and relevance components. The study also constructs the MVTec-LOCO-AD-Pair3C benchmark, framing IAD as a three-way classification task.

**Strengths:**

1. The work targets the efficiency problem in MLLM-AD research.
2. Experiments are comprehensively presented.

**Weaknesses:**

1. Generalization is questionable as training and evaluation are conducted on in-distribution data. The framework lacks validation on out-of-distribution scenarios.
2. The MVTec-LOCO-AD-Pair3C benchmark covers only five object categories, failing to comprehensively assess MLLM capabilities in diverse industrial settings. Additionally, the small test sample size raises concerns about the stability of the results, making the claimed performance advantages less credible due to potential fluctuations.
3. The proposed method lacks sufficient innovation. GPG and relevance reward are direct applications of existing reinforcement learning techniques. No novel mechanisms are introduced to address the unique challenges of AD tasks.

**Questions:**

The paper formulates industrial anomaly detection as a three-way classification task. Given the task’s relative simplicity compared to complex multimodal reasoning, could traditional supervised learning algorithms achieve competitive or even better performance?

---

> ### Author Response · Authors · 2025-11-24
>
> We thank the reviewer for their constructive critique. We have conducted additional experiments and analysis to address the concerns
>
>
> **Q1:** Concerns about Generalization (OOD validation).
>
> **A1:** We appreciate the reviewer pointing out the need for out-of-distribution (OOD) validation. To address this, we extended our evaluation to the MMAD benchmark, a large-scale, multi-domain dataset.
> The reasoins for chosing MMAD, that because it covers diverse industrial domains beyond the initial five categories, serving as a comprehensive testbed for MLLM-based anomaly detection generalization.
> We compared our method against a standard SFT baseline using the same backbone (Qwen2.5-VL-3B). The results are shown in **Table 1**.
>
> **Table 1. Evaluation on Target (FAD) and OOD (MMAD) Benchmarks.**
> | Model | FAD Acc (Target) | MMAD Avg | MMAD Prec | MMAD Recall | MMAD F1 |
> | :--- | :---: | :---: | :---: | :---: | :---: |
> | Baseline (Qwen2.5-VL-3B) | 39.77 | 63.3 | **72.3** | 34.5 | 40.8 |
> | SFT | 48.97 | 36.4 | 11.5 | 7.3 | 8.8 |
> | **Ours** | **55.89** | **63.9** | 71.9 | **40.3** | **44.7** |
>
> Analysis:
> The results clearly demonstrate our method's robustness. While standard SFT suffers from severe catastrophic forgetting (MMAD F1 drops to 8.8%), our method maintains strong generalization (MMAD F1 44.7%), outperforming both the Baseline and SFT. This confirms that our framework effectively generalizes to OOD scenarios without overfitting to the specific training categories.
>
> **Q2:** Concerns about innovation.
>
> **A2:** We respectfully argue that our innovation lies in addressing specific inefficiencies in current MLLM-based anomaly detection, rather than simply applying RL techniques:
>
> **Cost-Effective Training**: Current MLLM-based AD methods often require expensive data curation and high computational costs for fine-tuning. Our framework follows the GPG paradigm, offering a low-cost yet effective pathway to align general-purpose MLLMs with industrial needs, achieving significant performance gains with limited data overhead.
>
> **Enhancing Thinking Quality**: Recent works (e.g., [1], [2], [3]) often suffer from "hallucination" or "verbosity," where the model's reasoning process is overly lengthy and contains irrelevant information. To address this, we designed the Thinking Quality Reward. As shown in Fig. 3 in our paper, compared to the baseline, our method produces concise, logic-driven reasoning paths that are strictly relevant to the anomaly features. This mechanism effectively filters out noise and aligns the model's focus, representing a novel contribution to the IAD domain.
>
> **Q3:**  Comparison with Traditional Supervised Learning.
>
> **A3:**  We acknowledge that in terms of pure classification accuracy under the current setting, our method aims to achieve competitive performance comparable to traditional supervised learning algorithms.
>
> However, the key advantage of our approach lies in its capabilities beyond simple classification, which traditional methods cannot offer:
>
> **Interpretability (Why vs. What)**: Traditional models provide only a class label (e.g., "Class 1"). In contrast, our MLLM-based method outputs readable textual explanations (e.g., describing the defect type and reasoning). This semantic insight is critical for operators to understand the root cause of anomalies.
>
> **Open-World Adaptability**: Traditional models are limited to the specific defects they were trained on and require retraining for new categories. Our approach leverages the inherent knowledge of MLLMs, allowing it to generalize to unseen defect types and new domains (as validated by our MMAD results in A1) without parameter updates.
>
> [1] AnomalyR1: A GRPO-based End-to-end MLLM for Industrial Anomaly Detection
> [2] LAD-Reasoner: Tiny Multimodal Models are Good Reasoners for Logical Anomaly Detection
> [3] OmniAD: Detect and Understand Industrial Anomaly via Multimodal Reasoning

---

> > ### Comment · Reviewer_T6EL · 2025-11-27
> >
> > Thank you for the comments from the other reviewers on this paper and the authors' responses. However, I still hold a negative attitude towards this manuscript. While the proposed method may have some merits, it does not qualify as an insightful contribution. I recommend substantial revisions addressing the following points:
> >
> > ### **1. Generalization**
> >
> > The supplementary experiments provided in the rebuttal are insufficient to demonstrate the method's generalization ability. Its performance metrics on MMAD are relatively low, which is predictable given that FAD-TQ was only trained on partial MVTec-LOCO-AD data, leading to performance improvements limited to these specific categories. I suggest that the authors construct an additional untrained dataset containing logical anomalies, which can be curated from existing datasets such as VIADUCT [1] and MVTec [2]. This will better illustrate the proposed method's capability to distinguish between logical and structural anomalies.
> >
> > ### **2. Method Innovation**
> >
> > Although the authors emphasized the purposes of utilizing GPG and thinking quality reward in the rebuttal, these two designs are not novel in the field of reinforcement learning. The paper lacks a detailed analysis of the practical application of these improvements—such as their direct impacts, proof of working mechanisms, and performance when alternative approaches are adopted. Note that I do not require metrics from ablation studies, but rather in-depth analyses similar to those used in attention visualization.
> >
> > ### **3. Comparison with Traditional Supervised Learning**
> > I need to see the numerical upper bound achievable by traditional supervised models to facilitate the evaluation of MLLMs' performance. While I acknowledge the advantages of MLLMs highlighted in the rebuttal, the paper lacks an empirical evaluation of these claimed strengths. Furthermore, the logical anomaly tasks selected in this work inherently favor MLLMs. Traditional models are more sensitive to visual structural anomalies, whereas logical anomaly detection requires a certain level of reasoning ability, which large language models excel at.
> >
> > [1] Lehr, Jan, et al. "Ad3: Introducing a score for anomaly detection dataset difficulty assessment using viaduct dataset." European Conference on Computer Vision. Cham: Springer Nature Switzerland, 2024.
> >
> > [2] Bergmann, Paul, et al. "MVTec AD--A comprehensive real-world dataset for unsupervised anomaly detection." Proceedings of the IEEE/CVF conference on computer vision and pattern recognition. 2019.

---

### Official Review · Reviewer_YcV5 · 2025-10-31

**Soundness:** 3
**Presentation:** 3
**Contribution:** 3
**Rating:** 6
**Confidence:** 3

**Summary:**

This paper proposes the FAD TQ method, which fine tunes large multimodal models via reinforcement learning to perform image anomaly detection tasks while producing interpretable reasoning chains. FAD TQ leverages a Group Policy Gradient paradigm, eliminating the need for a reference model and KL divergence regularization, thereby directly optimizing the objective. The method introduces a two component reward function that balances reasoning chain efficiency and quality. Additionally, the authors propose a new evaluation method, MVTec LOCOAD Pair3C, based on existing datasets. Experimental results demonstrate that FAD TQ outperforms existing approaches in interpretability and accuracy.

**Strengths:**

S1.The use of Group Policy Gradient paradigm eliminates the reference model and KL divergence, which simplifies the RL process, reduces VRAM requirements, and speeds up training rollouts, making the approach more practical.
S2.Experimental results show that FAD-TQ improves accuracy by 16.12% over Qwen2.5 VL 3B and 17.05% over MiniCPM V 4, and achieves an 8.99% improvement over the GRPO training method, representing a substantial gain.

**Weaknesses:**

W1. The experimental comparison focuses only on accuracy; there is no evaluation of training cost or efficiency.
W2. The proposed FAD TQ model is trained solely on Qwen2.5 VL 3B, without testing the reinforcement learning method’s robustness on a wider range of base models.
W3. Only GRPO and GPG are used as baseline training methods; the comparison set is limited.

**Questions:**

1. Could experimental analysis be conducted to compare FAD TQ’s training cost against other reinforcement learning methods?
2. Could FAD TQ be trained on additional base models (e.g., Qwen3 VL Thinking with built in reasoning chains) to verify the robustness of the proposed training method?

---

### Official Review · Reviewer_b1hx · 2025-10-31

**Soundness:** 2
**Presentation:** 2
**Contribution:** 2
**Rating:** 2
**Confidence:** 4

**Summary:**

This paper introduces FAD-TQ, a new reinforcement learning (RL) framework for detailed industrial anomaly detection. The authors want to solve two common problems when using Multimodal Large Language Models (MLLMs) for this task: the reward signals are weak, and they rely too much on supervised fine-tuning (SFT) data. To tackle this, they propose a Group Policy Gradient (GPG) method and a new reward function called "Thought Quality" (TQ). They also created a new benchmark, MVTec-LOCO-AD-Pair3C, to show how their method improves MLLM performance.

**Strengths:**

+ The GPG algorithm is an RL method that doesn't need a reference model. This is great because it cuts down on computing costs and directly optimizes the main goal.
+ How the paper rewards both the "thinking process" and the final result separately. This seems like a really smart and necessary approach, especially for a task like anomaly detection.

**Weaknesses:**

+ A major concern is the results on their new benchmark. Even with test data and powerful MLLMs, their method doesn't actually perform better than simpler, smaller models that just do a basic "yes/no" classification. This makes me seriously question if their complicated approach is even necessary.
+ The paper argues that GPG is a great way to unlock the full potential of MLLMs. However, they don't compare it against a standard SFT approach. Without that comparison, it's hard to tell if GPG is truly needed.
+ The paper tries to turn anomaly detection into a more detailed "anomaly classification." The benefit of doing this isn't clear. They should probably also include the overall anomaly detection accuracy (just a simple "is it an anomaly or not?") as a baseline metric to help us judge their results better.

**Questions:**

Please refer to the **weakness** section.

---

> ### Author Response · Authors · 2025-11-20
>
> We thank the reviewer for this thoughtful observation. Our responses are as follows. We will incorporate them after revision.
>
> **Q1:** Necessity of the Approach for the fine-grained anomaly detection.
>
> **A1:** We acknowledge that simpler models may achieve high performance on binary anomaly detection tasks (Normal v.s. Anomalous).
> However, we respectfully argue that the value of our method extends beyond binary classification and addresses a critical gap in industrial anomaly understanding, including providing a readable reasoning process.
>
> - Extending the Frontier of IAD: While binary detection is mature, we believe the field must evolve from simple "detection" to "diagnosis and reasoning." Our work explores the potential of MLLMs to perform multi-class classification (e.g., distinguishing between logical and structural anomalies), which simple binary classifiers cannot achieve.
>
> - Practical Necessity in Industrial Settings: In real-world manufacturing, detecting an anomaly is only the first step. Structural anomalies (e.g., scratches, stains) and Logical anomalies (e.g., missing components, wrong placement) require entirely different downstream responses:
>
>   - Structural defects (e.g., scratches, stains) often indicate process instabilities. They typically require production line adjustments or product scrapping.
>   - Logical defects (e.g., missing components, wrong placement) usually arise from assembly errors, demanding immediate intervention or distinct re-routing strategies.
>
> The remedial costs and handling pipelines for these two types differ significantly. Therefore, a simple "yes/no" output is insufficient for automated decision-making in modern factories.
>
> - The Role of Complexity: The complexity of our approach is not intended to merely improve binary accuracy, but to unlock the capability for fine-grained categorization and semantic understanding. Our work serves as a proactive step to future-proof industrial AI agents against increasingly complex manufacturing scenarios.
>
> **Q2:** Comparison with standard SFT to validate the necessity of our approach.
>
> **A2:** We have implemented a standard SFT pipeline using the same base model (Qwen2.5-VL-3B) to validate the necessity of our approach. The comparison results are reported in **Table 1**.
>
> **Table 1. Comparison with Standard SFT.** *FAD denotes our Fine-grained Anomaly Detection benchmark, and MMAD denotes the large-scale benchmark under one-shot setting.*
>
> | Model | FAD Acc | MMAD Avg | MMAD Prec | MMAD Recall | MMAD F1 |
> | :--- | :---: | :---: | :---: | :---: | :---: |
> | Baseline (Qwen2.5-VL-3B) | 39.77 | 63.3 | **72.3** | 34.5 | 40.8 |
> | SFT | 48.97 | 36.4 | 11.5 | 7.3 | 8.8 |
> | **Ours** | **55.89** | **63.9** | 71.9 | **40.3** | **44.7** |
>
> **Analysis:**
>
> To comprehensively validate the effectiveness of our approach, we conducted additional evaluations on the large-scale MMAD benchmark, which represents a significant domain shift from our training data.
>
> 1. **Catastrophic Forgetting in Standard SFT:** Standard SFT exhibits severe **catastrophic forgetting** on the MMAD benchmark. While it improves FAD Accuracy to 48.97%, it drastically degrades generalization performance—MMAD Average drops to 36.4%, with Recall plummeting to **7.3%** and F1 to **8.8%**. This demonstrates that naive fine-tuning fails to preserve the model's general-purpose capabilities.
>
> 2. **Robust Generalization of Our Method:** In stark contrast, our method achieves **superior performance on both benchmarks simultaneously**. On the target FAD task, we reach the highest Accuracy (**55.89%**, +6.92% over SFT, +16.12% over Baseline). Critically, on the out-of-distribution MMAD benchmark, our method maintains robust generalization with an Average of **63.9%** (comparable to the Baseline's 63.3%), while significantly improving Recall to **40.3%** (+5.8%) and F1 to **44.7%** (+3.9%).
>
> 3. **Key Advantage:** Our approach successfully addresses the fundamental trade-off between task-specific adaptation and general capability preservation. Unlike standard SFT, which sacrifices generalization for target task performance, our method achieves good results, substantial improvement on the fine-grained detection task while maintaining strong cross-domain robustness. This validates that our design effectively prevents overfitting and enables practical deployment in diverse industrial scenarios.

---

> > ### Author Response · Authors · 2025-11-20
> >
> > **Q3:** Justification for anomaly classification and binary performance comparison.
> >
> > **A3:** Under the binary setting, we implemented the baseline model (Qwen2.5-VL-3B), its SFT variant, and our method. The comparison results are reported in Table 2.
> >
> > **Table 2. Binary Setting Comparison on FAD Benchmark.**
> >
> > | Model | Accuracy | Precision | Recall | F1 |
> > | :--- | :---: | :---: | :---: | :---: |
> > | Baseline (Qwen2.5-VL-3B) | 38.0 | 50.8 | **100.0** | **67.4** |
> > | SFT | 60.4 | 48.5 | 34.7 | 40.5 |
> > | **Ours** | **63.7** | **60.4** | 69.9 | **64.8** |
> >
> > **Analysis:**
> >
> > 1. **Failure of Baselines:** Although the Baseline achieves a high Recall (100.0%) and consequently a high F1 (67.4), these metrics are misleading.The low Accuracy (38.0%) reveals that the baseline essentially acts as a trivial predictor (classifying almost everything as anomalous), lacking real discriminative ability.
> > 2. **Limitations of Standard SFT:** Standard SFT shows improvement in Accuracy (60.4%) but becomes overly conservative, resulting in poor Recall (34.7%) and F1 score (40.5%). This suggests that SFT struggles to balance precision and recall in the binary setting.
> > 3. **Effectiveness of Ours:** Our method achieves the best overall performance, reaching the highest **Accuracy (63.7%)** and **Precision (60.4%)**, while maintaining competitive Recall (69.9%) and **F1 score (64.8%)**. This demonstrates that our approach successfully balances discriminative ability and sensitivity to anomalies in the binary detection task.

---

> > > ### Comment · Reviewer_b1hx · 2025-11-24
> > >
> > > Thanks for the authors' response. I'm willing to improve my rating, but I think the paper needs extensive revisions before it can be accepted. The comparison between SFT and GRPO makes sense. However, I slightly disagree with A1. There are multiple papers about anomaly type classification[1,2]. Though they didn't conduct experiments on MVTec-LOCO, We can see that the abnormal binary and multi-class classifications are not unrelated. Especially in LLMs, their understanding of binary and multi-class anomalies should be mutually reinforcing. If LLMs have a promising performance in multi-class defect classification, they should work well in binary anomaly classification too. What's more, GRPO usually used to align LLMs' thinking path with human, but it doesn't inject knowledge well, but Qwen doesn't have too much knowledge about anomaly detection, maybe this is the reason why FAD-TQ didn't get a high acc score. To make the paper more acceptable, I think there are still several points need to be improved. The paper should identify why SFT performs worse than GRPO after anomalous knowledge injection, and it should consider how to simultaneously improve the performance of binary defect detection and multi-class defect classification.
> > >
> > > [1] MVREC: A General Few-shot Defect Classification Model Using Multi-View Region-Context
> > > [2] UniADC: A Unified Framework for Anomaly Detection and Classification

---

### Meta-Review · Area_Chair_VwU6 · 2026-01-06

**Summary:**

The paper explores using LLMs (specifically comparing SFT and GRPO) for anomaly detection tasks.
The review process resulted in a mixed assessment (Scores: 2, 6, 4, 4). While one reviewer saw potential, the majority expressed concerns regarding the depth of the analysis and the novelty of the contribution. Despite the authors' rebuttal, the critical reviewers maintained that the paper requires extensive revisions regarding its methodological premises and experimental analysis before it is ready for publication. The decision is Reject.

**Reviewer Concerns:**

1)  A major outstanding concern involves the relationship between binary anomaly detection and multi-class defect classification. The reviewer argued that these tasks should be mutually reinforcing in LLMs, countering the authors' response (A1). Current maniscript fails to adequately explore why the model (Qwen) struggles to leverage multi-class performance for binary detection.
2) Multiple reviewers maintained that the overall novelty and contribution remain insufficient compared to existing methods

**Reviewer Scores:**

Since the required improvements (simultaneously improving binary/multi-class performance, deep analysis of knowledge injection vs. alignment) go beyond the scope of a camera-ready update, the paper is rejected.

---

### Decision · Program_Chairs · 2026-01-26

Reject